# Seasonal variations in respiratory morbidity in primary care and its correlation with the quality of air in urban Odisha, India

**Abhinav Sinha**◉, **Jitendriya Amrit Pritam**◉, **Hitesh Kumar Jain, Sidhartha Giri, Sanghamitra Pati, Jaya Singh Kshatri** *

ICMR-Regional Medical Research Centre, Bhubaneswar, Odisha, India

◉ These authors contributed equally to this work.
* jsk.icmr@outlook.com

## Abstract

Poor air quality, especially in urban regions among low-and middle-income countries such as India poses a significant healthcare challenge. Amongst urban areas, metropolitan cities garner the utmost importance for air quality related policies and studies with limited studies from tier II cities which are thought to be relatively immune to air pollution. Hence, we aimed to identify the most frequent respiratory morbidities and explore its correlation with exposure to ambient PM2.5 particles in Bhubaneswar (a tier II city in coastal India), Odisha. A chart review was carried out through data extracted from the records of urban health centres. Data on PM2.5 concentrations were obtained from Odisha State Pollution Control Board. The morbidities were coded by using the International Classification of Primary Care-2 system (ICPC-2). Descriptive statistics such as incidence of respiratory illnesses was computed across seasons. The ecological correlation between respiratory morbidity patterns and corresponding concentration of PM2.5 in air was analysed for each season. A positive correlation (r = o.94) between PM2.5 and respiratory morbidities was observed. The incidence of respiratory morbidities was 183.31 per 1000 person year. We identified 21 out of 43 respiratory diseases classified under ICPC-2. Upper Respiratory Tract Infection was the most commonly (116.8 per 1000 person year) incident condition. We observed one-fourth increase in the incidence of respiratory illnesses during winters. Respiratory morbidities are common in urban Bhubaneswar which follows a seasonal pattern and are possibly linked with the seasonal variations in levels of PM2.5 particles. Our study highlights that tier II cities are equally prone to health effects of air pollution. Future programmes and policies should take these cities into consideration too.

## Introduction

Respiratory morbidities continue to rise worldwide posing a significant challenge for the healthcare systems [1]. This becomes severe among people living in low-and-middle income countries (LMICs) where quality of air is usually compromised owing to indoor and outdoor pollution, urbanization and industrialization [2]. Additionally, LMICs lack resources for

**Data Availability Statement:** The data used in the study were obtained from two sources i.e. Primary Healthcare Centres (PHCs) of Bhubaneswar, and Odisha State Pollution Control Board (OSPCB). The

data is solely the property of Government of Odisha, given to the authors for conducting particularly this study. The data analysed during the current study can either be obtained directly from PHCs of Odisha (through permission of State Research and Ethics Committee, Directorate of Health Services, Govt. of Odisha: https://dhsodisha. nic.in/) and OSPCB (by requesting them directly: https://ospcboard.org/contact-us/) or from corresponding author on reasonable request.

**Funding:** The authors received no specific funding for this work.

**Competing interests:** The authors havSe declared that no competing interests exist.

monitoring and maintaining air quality [3]. According to World Health Organization (WHO), the quality of air has an adverse effect on the quality of life, and both are deteriorating day by day [4]. A similar situation is seen in India where rapid urbanization, industrialization, high levels of energy utilization, increased number of motor vehicles and use of biomass fuel has led to increased respiratory morbidities and associated mortality [5]. Estimates suggest that the high levels of particulate matter of size <2.5μm (PM2.5) can potentially jeopardize lung function as they enter deep into the lungs thus, impairing alveolar wall [6]. Additionally, there is evidence of a seasonal variation in size as well as concentration of this particulate matter which in turn can lead to a seasonal variation in incidence of respiratory illnesses [7]. The GBD estimates suggest that the exposure to PM2.5 particles are responsible for 59% of total deaths and disability adjusted life years (DALYs) in East and South Asia [8]. Air pollution causes and exacerbates chronic respiratory diseases such as asthma, chronic obstructive pulmonary disease (COPD), and emphysema [9]. The prevalence of Chronic Obstructive Pulmonary Disease was reported to be 4.2% (95% UI: 4.0–4.4%) while that of asthma was 2.9% (95% UI: 2.7–3.1%) as reported by the Global Burden of Diseases (GBD), 2016 [10].

India has taken several steps to combat air pollution since Air (Prevention and Control of Pollution) Act, 1981; Air (Prevention & Control of Pollution) rules, 1942; National Ambient Air Quality Standards, 1982; Air (Prevention & Control of Pollution) (Union Territories) Rules; 1983; Environment (Protection) Act, 1986; Air (Prevention & Control of Pollution) Amendment Act; 1987; Motor Vehicles Act, 1988; National Environment Tribunal Act, 1995; National Environmental Appellate Authority Act, 1997; Revised Air Quality, 1994; Environmental Impact Assessment Notification, 1994; Environment Pollution Control Authority (EPCA), 1998; Environmental Impact Notification, 2006; National Ambient Air Quality Standards, 2009; National Green Tribunal Act, 2010; Graded Response Action Plan (GRAP), 2017; Motor Vehicles Act, 2019; National Clean Air Programme, 2019; Commission for Air Quality Management, 2020 to monitor and report real time statistics of air quality [11, 12]. Additionally, India signed Paris Accord in 2016 [13]. Moreover, the recent schemes such as Ujjwala Yojana provides free clean cooking fuel to poor households with an aim to reduce ambient air pollution (AAP) and household air pollution (HAP) throughout the country [14, 15]. However, most of the studies documenting the association of respiratory illnesses and air quality are restricted to either national capital region [16, 17] or metropolitan cities [18] with no studies reported from Bhubaneswar, Odisha. Bhubaneswar is one of the most rapidly urbanizing cities in India and the projected population growth is also high. Hence, having a better understanding of the health effects would not only help the National efforts, but would be essential for framing state level efforts [19]. While this example is for Bhubaneswar but such coastal cities in the South and South-east Asian region with similar or near similar challenges would also be benefitted from such a study [20].

Hence, we undertook a chart review to identify the most frequent respiratory morbidities encountered in primary care and examine their seasonal variations in Bhubaneswar, Odisha. Additionally, we aimed to explore the correlation between burden of respiratory illnesses and exposure to ambient PM2.5 particles.

## Materials & methods

### Study design, setting and population

A retrospective observational study was carried out at the urban primary healthcare centres (UPHC) of Bhubaneswar, the capital of Odisha. Odisha, a coastal state in eastern India has a tropical climate characterized by high temperature, high humidity, medium to high rainfall

and short and mild winters (Climate) [21]. Additionally, the state has many industrial clusters predominantly made up of heavy industries and mining (Industries) [22]. This makes Odisha vulnerable to varied levels of air pollution in different seasons and hence, variations in respiratory morbidities may be reported [23]. A previous study conducted in Bhubaneswar reported that the concentration of PM2.5 particles were about 3.5 times higher than in pre-monsoon which could be attributed to predominant coarse particles originating from long range transport of pollutants from northern and western India and from West Asia as well [24]. Moreover, the characteristics of PM2.5 particles in winter season show that the mean PM2.5 was in the range of 101–142μg/m$^3$ which is 65–135% higher than the 24 hour average National Ambient Air Quality Standards; spectrometric measurement depicted a diurnal pattern with morning and evening peaks [24]. The major sources of PM2.5 were vehicular emissions, industrial, combustion, and crustal sources [25]. The city was divided into four geographical zones east, west, north and south for this study. Four out of twenty nine Urban Primary Health Centre (UPHC) and Urban Community Health Centre (UCHC) of Bhubaneswar were selected randomly, one from each zone. Each PHC caters to the needs of 30,000 populations and serves a similar patient load per month. Data of all the patients who attended these clinics in the previous year were obtained from the existing out-patient department (OPD) registers. Patients whose complete details were not available in the OPD register were excluded. Also, we excluded data if the OPD register was torn or the handwriting was not clearly visible. This study was reported following Strengthening the reporting of observational studies in epidemiology (STROBE) (S1 Checklist).

## Data collection

**Health data.**   At each UPHC, a standard register is maintained by either the staff nurse or the physician to register the records of patients. The information on age, sex, chief complaint and clinician's diagnosis were extracted from these records. Data were collected from January to December in the 2018 and was classified based on the major seasons according to local geo-climatic conditions. This constituted of rainy season (16 weeks) from July to October, winter season (16 weeks) from November to February and summer season (20 weeks) from March to June. In this manner, data of 10, 900 patients were extracted in total. The data were first extracted based on month of clinic visit and later on classified into seasons at analysis. These data were entered into MS Excel and coded by using the International Classification of Primary Care-2 (ICPC-2) system which is widely utilized to classify primary care encounters [26]. All respiratory morbidities were segregated as per the ICPC-2 system.

**Air quality data.**   Data on the monthly average values of PM2.5 concentrations were obtained from six pollution monitoring stations of Odisha State Pollution Control Board (SPCB) located in Bhubaneswar namely SPCB Office Building, Unit-8; IRC Village, Nayapalli; Capital Police Station, Unit-1; Palasuni; Patrapada; and Chandrasekharpur. The SPCB follows the guidelines for sampling and measurement of notified Ambient Air Quality Parameters (NAAQS), where PM2.5 is monitored 24 hourly which is compiled with 98% of the time in a year. 2% of the time, they may exceed the limits but not on two consecutive days of monitoring. It is measured with Gravimetric method following all quality control and quality assurance measures. All activities are recorded in the field log book during sampling which includes sites name, filter ID, sample start and stop dates and time and filed operator initials. Subsequently, the monthly average data of PM2.5 were then converted to the mean seasonal data by dividing the sum of monthly average PM.5 values of the corresponding months by number of months considered for each season.

## Data management and statistical analysis

The data were analysed using STATA v 16.0 (STATA Corps, Texas). This study intended to report a chart review of respiratory morbidities and its correlation with air pollution and hence, only descriptive statistics were used. Frequency and proportions were calculated for age, sex, chief complaint and clinician's diagnosis along with its seasonal variations. The respiratory morbidities were presented as incident cases per 1000 population year with regard to ICPC code of respiratory diseases. We used graphs and Pearson Correlation Coefficient to depict the correlation between incident respiratory morbidity variations and corresponding concentration of particulate matter PM2.5 in air. Sub-group analysis based on age, sex and seasonal variation was done. The incidence of morbidities was calculated using the following formula: (new cases) / (population x timeframe) per 1000 population [27]. Since, our aim was to calculate the incidence of respiratory morbidities presenting at the selected facilities, we took the total number of cases present at the facilities as the denominator (population).

## Ethical considerations

This study was approved by the Institutional Human Ethical Committee of ICMR-Regional Medical Research Centre, Bhubaneswar. Permission for data collection was obtained from district authorities. Written consent from participants was not required as anonymous data were extracted from OPD registers. Additionally, authors did not had access to information such as contact number and full address that could identify individual participants during or after data collection as the registers maintained by the clinician have name, age, sex, provisional diagnosis and treatment given.

## Results

The overall incidence of respiratory morbidities was 183.31 per 1000 person year. Further analysis was done on the 1998 (18.3%) patients who had respiratory illnesses out of 10900 extracted records. A majority of the study sample comprised of males (58.9%). We observed the highest incidence of respiratory illnesses in males among all age groups except infants <1 year. Amongst participants aged 18–60 years, the incidence of respiratory illnesses was 46.97 per 1000 person year in females whereas 69.44 among males. The distribution of respiratory illness across different age groups and gender is presented in Fig 1.

The distribution of respiratory illnesses according to ICPC-2 classification and its incidence across age groups is presented in Table 1. A total 43 respiratory diseases are classified under ICPC-2 system of disease classification, but we found only 21 diseases among participants of this study. We observed upper respiratory tract infection, URTI (R-74) to be the most common encountered condition in the selected facilities. The incidence of URTI was 76.79 per 1000 person year among adults. Other most frequently encountered respiratory conditions were cough (R05) with an incidence of 14.5 per 1000 person year and tonsillitis acute (R76) with an incidence of 9.81 per 1000 year. However, sneezing/nasal congestion (R07), strep throat (R72) and boil/abscess nose (R73) comprised the minimum reported diseases.

Respiratory conditions followed a seasonal pattern where incidence increased from 53.11 per 1000 year during summer to 54.67 per 1000 person year during rainy season. Further, it increased to 75.5 per 1000 person year during winter season. A high incidence of URTI (51.56 per 1000 person year), cough (5.41 per 1000 person year) and COPD (2.84 per 1000 person year) was recorded during winter season. Tonsillitis acute (4.86 per 1000 person year) was seen mostly during rainy season while respiratory infection other (6.42 per 1000 person year) was observed more during summer. The detailed incidence of respiratory conditions across various seasons is presented in Table 2.

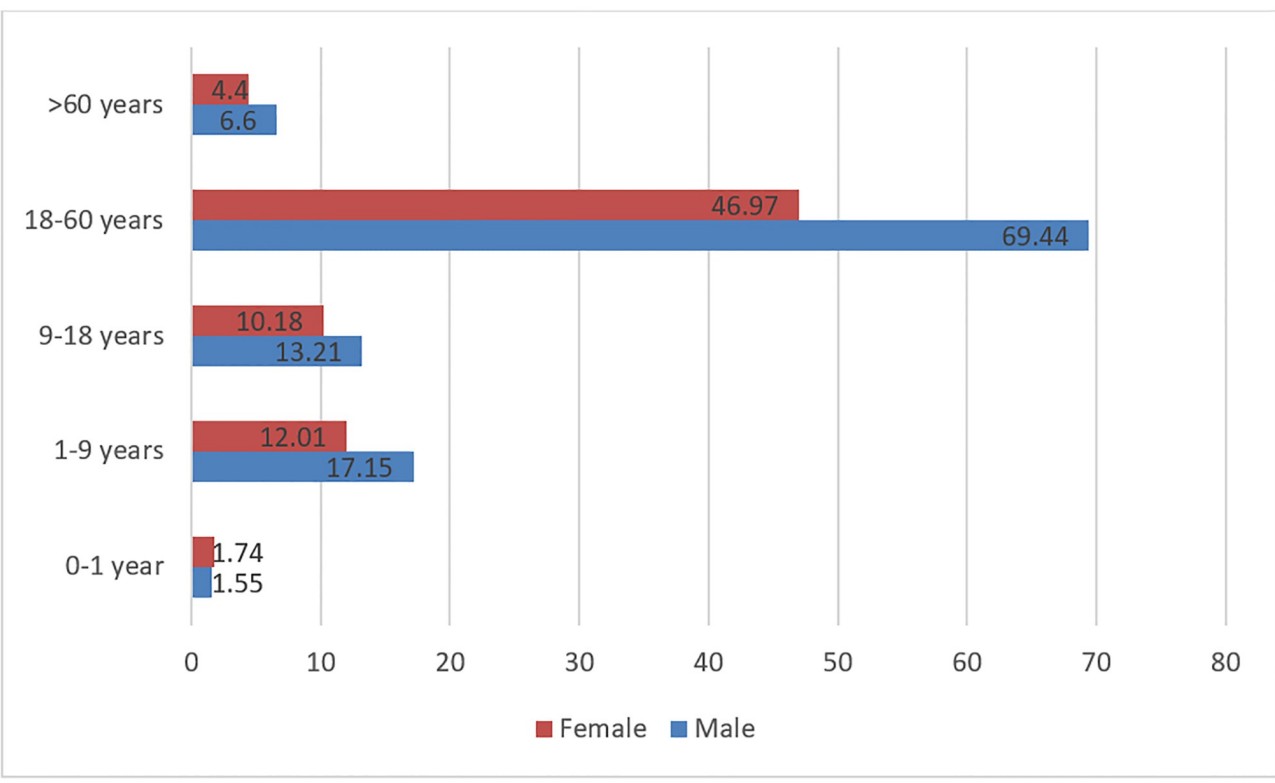

**Fig 1. Distribution of respiratory diseases across demographic characteristics.**

The trend of monthly average levels of PM2.5 particles is shown in Fig 2. A positive correlation between PM2.5 and incident respiratory morbidities was observed in the study area (Fig 3). The Pearson correlation coefficient between PM2.5 and incident respiratory morbidities was r = 0.94.

## Discussion

We observed the incidence of respiratory diseases to be around 183 per 1000 person year which is consistent with the findings of our previous chart review reporting morbidity patterns among adults attending primary care in Odisha where we found respiratory morbidities were the most common encountered condition [28]. Here, it is worth noting that the reporting of respiratory morbidities has increased between 2014 and 2018 in the area. Additionally, we observed the highest incidence of respiratory illnesses was among adults which indicated this disease to be akin amongst the productive group, a potential cause for absenteeism and economic loss.

Further, we classified respiratory illnesses based on ICPC-2 system which enabled us to easily comprehend the diseases [29]. Since, various terminologies were used to report the symptoms or provisional physician's diagnosis in the records, coding helped us in maintaining the uniformity. Aligning with our study, previous studies documented ICPC-2 classification to be a simple yet practically useful guide for primary care research [30, 31]. We observed URTI and cough to be widely reported in our study which is consistent with the findings of a similar study from India [32]. Interestingly, we did not observe much difference in the distribution of morbid conditions across gender as the reported conditions were almost in similar proportion

**Table 1. ICPC coding of respiratory diseases across age groups.**

| Sl. No | Name of the disease | ICPC code | Total (n) | Infant | | Child | | Adolescent | | Adult | | Elderly | |
|---|---|---|---|---|---|---|---|---|---|---|---|---|---|
| | | | | n | Incidence | n | Incidence | n | Incidence | n | Incidence | n | Incidence |
| 1 | Pain Respiratory system | R01 | 56 | 1 | 0.09 | 3 | 0.28 | 4 | 0.37 | 43 | 3.94 | 5 | 0.46 |
| 2 | Breathing problem, other | R04 | 3 | 0 | 0.00 | 2 | 0.18 | 0 | 0.00 | 1 | 0.09 | 0 | 0.00 |
| 3 | Cough | R05 | 158 | 2 | 0.18 | 21 | 1.93 | 27 | 2.48 | 99 | 9.08 | 9 | 0.83 |
| 4 | Nose bleed/ epistaxis | R06 | 4 | 0 | 0.00 | 1 | 0.09 | 0 | 0.00 | 3 | 0.28 | 0 | 0.00 |
| 5 | Sneezing/ nasal congestion | R07 | 1 | 0 | 0.00 | 1 | 0.09 | 0 | 0.00 | 0 | 0.00 | 0 | 0.00 |
| 6 | Nose symptom | R08 | 7 | 0 | 0.00 | 1 | 0.09 | 0 | 0.00 | 6 | 0.55 | 0 | 0.00 |
| 7 | Sinus symptom | R09 | 3 | 0 | 0.00 | 0 | 0.00 | 0 | 0.00 | 3 | 0.28 | 0 | 0.00 |
| 8 | Throat symptom | R21 | 34 | 0 | 0.00 | 4 | 0.37 | 7 | 0.64 | 21 | 1.93 | 2 | 0.18 |
| 9 | Respiratory symptom | R29 | 7 | 0 | 0.00 | 0 | 0.00 | 1 | 0.09 | 6 | 0.55 | 0 | 0.00 |
| 10 | Strep throat | R72 | 1 | 0 | 0.00 | 0 | 0.00 | 1 | 0.09 | 0 | 0.00 | 0 | 0.00 |
| 11 | Boil / abscess nose | R73 | 1 | 0 | 0.00 | 0 | 0.00 | 0 | 0.00 | 1 | 0.09 | 0 | 0.00 |
| 12 | URTI | R74 | 1273 | 26 | 2.39 | 199 | 18.26 | 149 | 13.67 | 837 | 76.79 | 62 | 5.69 |
| 13 | Sinusitis acute/ chronic | R75 | 13 | 0 | 0.00 | 1 | 0.09 | 2 | 0.18 | 9 | 0.83 | 1 | 0.09 |
| 14 | Tonsillitis acute | R76 | 107 | 0 | 0.00 | 28 | 2.57 | 31 | 2.84 | 46 | 4.22 | 2 | 0.18 |
| 15 | Acute bronchitis/ bronchiolitis | R78 | 38 | 0 | 0.00 | 2 | 0.18 | 4 | 0.37 | 26 | 2.39 | 6 | 0.55 |
| 16 | Chronic bronchitis | R79 | 4 | 0 | 0.00 | 1 | 0.09 | 1 | 0.09 | 1 | 0.09 | 1 | 0.09 |
| 17 | Pneumonia | R81 | 5 | 0 | 0.00 | 0 | 0.00 | 1 | 0.09 | 4 | 0.37 | 0 | 0.00 |
| 18 | Respiratory infection other | R83 | 168 | 7 | 0.64 | 44 | 4.04 | 17 | 1.56 | 90 | 8.26 | 10 | 0.92 |
| 19 | Chronic obstructive pulmonary dis | R95 | 56 | 0 | 0.00 | 2 | 0.18 | 3 | 0.28 | 35 | 3.21 | 16 | 1.47 |
| 20 | Asthma | R96 | 52 | 0 | 0.00 | 5 | 0.46 | 7 | 0.64 | 36 | 3.30 | 4 | 0.37 |
| 21 | Allergic rhinitis | R97 | 7 | 0 | 0.00 | 3 | 0.28 | 0 | 0.00 | 2 | 0.18 | 2 | 0.18 |
| | | Total | 1998 | 36 | 3.3 | 318 | 29.18 | 255 | 23.39 | 1269 | 116.43 | 120 | 11.01 |

**Table 2. Seasonal variations of respiratory morbidities.**

| Respiratory Disease | ICPC code | Total | Rainy | | Summer | | Winter | |
|---|---|---|---|---|---|---|---|---|
| | | | n | Incidence | n | Incidence | n | Incidence |
| Pain Respiratory system | R01 | 56 | 16 | 1.47 | 26 | 2.39 | 14 | 1.28 |
| Breathing problem, other | R04 | 3 | 0 | 0 | 3 | 0.28 | 0 | 0 |
| Cough | R05 | 158 | 51 | 4.68 | 48 | 4.4 | 59 | 5.41 |
| Nose bleed/ epistaxis | R06 | 4 | 1 | 0.09 | 1 | 0.09 | 2 | 0.18 |
| Sneezing/ nasal congestion | R07 | 1 | 1 | 0.09 | 0 | 0 | 0 | 0 |
| Nose symptom | R08 | 7 | 1 | 0.09 | 1 | 0.09 | 5 | 0.46 |
| Sinus symptom | R09 | 3 | 0 | 0 | 2 | 0.18 | 1 | 0.09 |
| Throat symptom | R21 | 34 | 2 | 0.18 | 16 | 1.47 | 16 | 1.47 |
| Respiratory symptom | R29 | 7 | 0 | 0 | 6 | 0.55 | 1 | 0.09 |
| strep throat | R72 | 1 | 1 | 0.09 | 0 | 0 | 0 | 0 |
| Boil / abscess nose | R73 | 1 | 0 | 0 | 1 | 0.09 | 0 | 0 |
| URTI | R74 | 1273 | 401 | 36.79 | 310 | 28.44 | 562 | 51.56 |
| Sinusitis acute/ chronic | R75 | 13 | 3 | 0.28 | 1 | 0.09 | 9 | 0.83 |
| Tonsillitis acute | R76 | 107 | 53 | 4.86 | 36 | 3.3 | 18 | 1.65 |
| Acute bronchitis/ bronchiolitis | R78 | 38 | 9 | 0.83 | 14 | 1.28 | 15 | 1.38 |
| Chronic bronchitis | R79 | 4 | 4 | 0.37 | 0 | 0 | 0 | 0 |
| Pneumonia | R81 | 5 | 0 | 0 | 5 | 0.46 | 0 | 0 |
| Respiratory infection other | R83 | 168 | 31 | 2.84 | 70 | 6.42 | 67 | 6.15 |
| Chronic obstructive pulmonary disease | R95 | 56 | 9 | 0.83 | 16 | 1.47 | 31 | 2.84 |
| Asthma | R96 | 52 | 13 | 1.19 | 16 | 1.47 | 23 | 2.11 |
| Allergic rhinitis | R97 | 7 | 0 | 0 | 7 | 0.64 | 0 | 0 |
| **Total** | | **1998** | **596** | **54.67** | **579** | **53.11** | **823** | **75.5** |

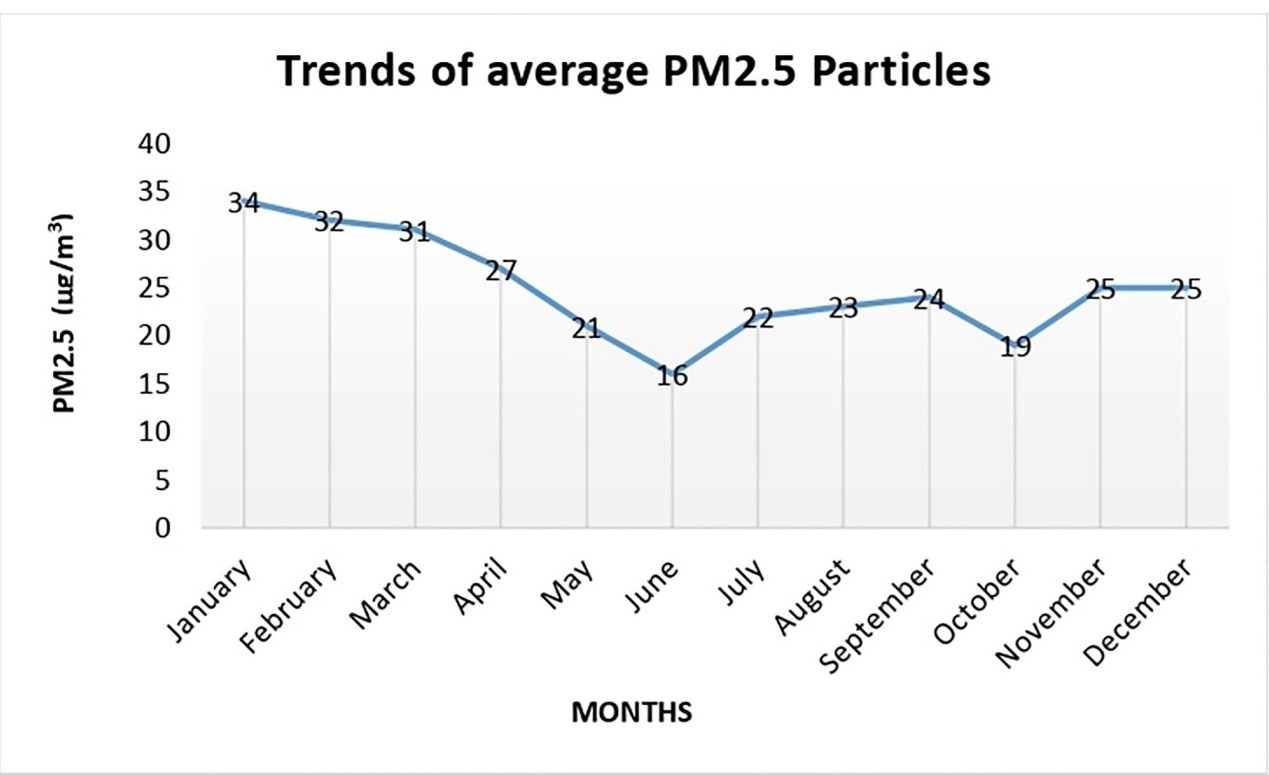

**Fig 2. Trends of monthly average levels of PM2.5 particles.**

between the two groups. This further exhibits a positive indication that there is no difference in the healthcare seeking behaviour across gender.

We observed incidence of respiratory morbidities increased during rainy season and were at peak during winters. Our findings correlate with the results of a previous study which reported a threefold increase in respiratory morbidities during winter [33]. This could be attributed to rise in viral infections such as influenza virus, rhinoviruses etc., and finer size of PM2.5 granules during winter. Furthermore, our findings show, respiratory illnesses increased with a rise in PM2.5 levels in air which also corresponds to the winter season. Deteriorated air quality has long been associated with respiratory illnesses as shown by various previous studies [6, 34, 35]. Similar to our findings, a previous review estimated that PM2.5 particles contributed to a 6716 (95% CI: 3639–9482) cases of respiratory disease in Southeast Asia [36]. Another systematic review of cohort studies conducted in Asia-Pacific region showed that long-term exposure to PM2.5 particles increased the incidence of cardiovascular diseases, kidney diseases, type 2 diabetes mellitus, and chronic obstructive pulmonary disease [37]. A recent review of time-series analysis revealed that the association between PM2.5 and health effects was significantly stronger for respiratory diseases (for short term studies i.e. less than 7 years) as compared to cardiovascular diseases [38]. This trend was commonly observed among metropolitan cities with greater extent of urbanization, vehicular emission and on-going construction work [39] but, its correlation in our study setting (a tier-II city) is a serious concern. Our study findings exhibits that the effects of air pollution are no longer confined to metros but have also gripped other parts of country, such as smaller cities, which were considered to be relatively immune. Although, the recent programmes such as National Clean Air

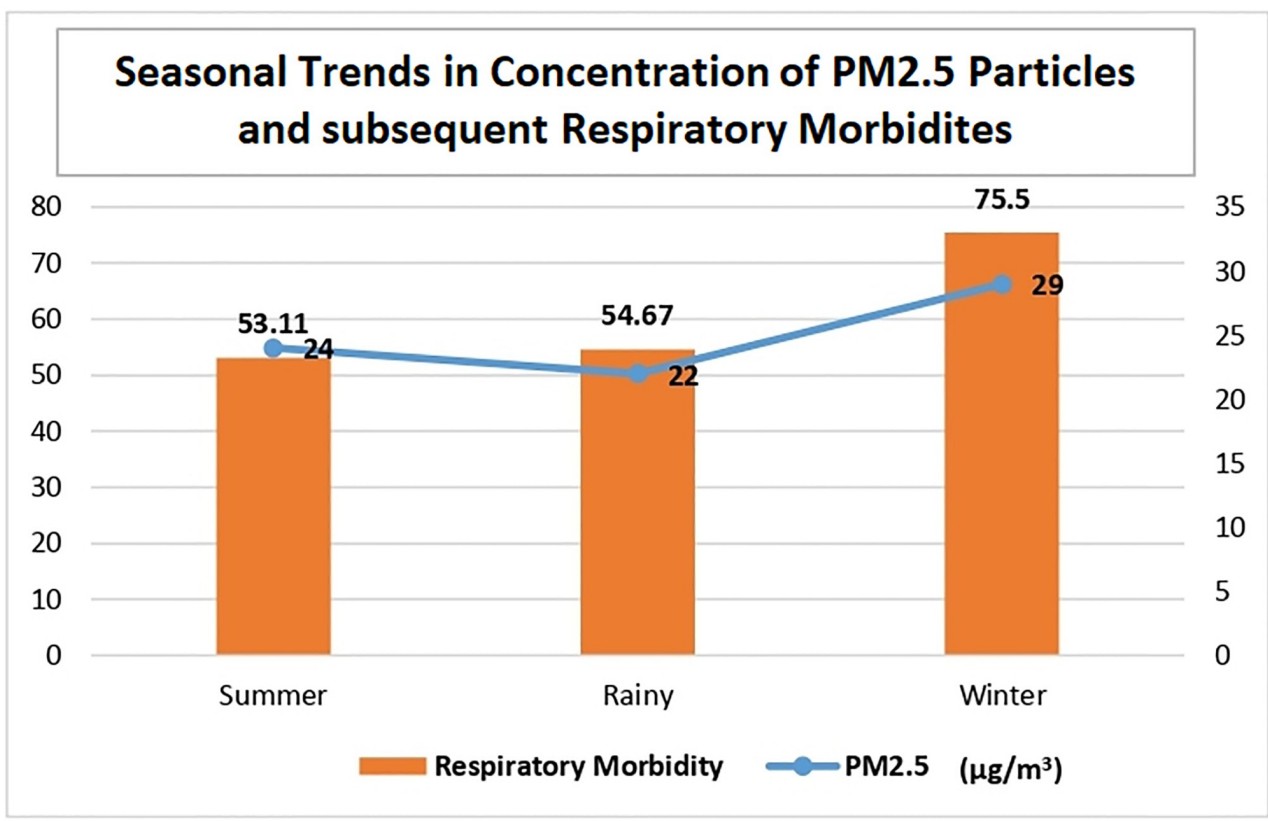

**Fig 3. Seasonal trends in concentration of PM2.5 particles and subsequent respiratory morbidities.**

Programme (NCAP), 2019 targets to reduce 20–30% of air pollution at national level by 2024, still more studies are warranted from non-metropolitan cities to garner evidence for future policies.

The finding of this study draws attention towards the respiratory morbidities in primary care and its correlation with air pollution. It is evident that respiratory morbidities are evidently correlated with the air pollution which needs attention at all levels. There is an urgent need for clean fuels and stringent emission norms for vehicles to reduce air pollution. Alternately, public transport should be scaled up so as to restrain the use of personal vehicles. For long distances, car pool should also be considered. Cycling should be promoted by making safe cycle tracks along with pedestrian tracks. Power plants should be located on outskirts of cities and emissions from them needs to be checked. Pollution from other industries should also be controlled through proper implementation of rules and regulations. Combustion of garbage, crop fires, biomass based cook stoves etc. should be eliminated at the earliest. Primary care should be strengthened for timely, accessible and egalitarian provision of services which can be achieved through increasing Health and Wellness Centres (HWCs) which envisage providing a package of preventive and curative services. Future studies are warranted to establish causality.

This is the first study to explore seasonal variations in respiratory diseases and link it with the quality of air in the Eastern part of India. Additionally, we coded morbidities based on ICPC-2 classification system which helped in standardizing patient reported symptoms based diagnosis in primary care. This study being cross-sectional in nature could not establish

causality. Our study is also limited by the use of record based data which included provisional diagnosis as certain conditions such as COPD are difficult to diagnose without investigations and the final diagnosis might have changed subsequently. Moreover, we did not adjust for any other physical/physiological aspects of the patients in this study.

## Conclusion

This chart review suggests respiratory morbidities to be common in primary care which shows seasonal variation. Additionally, deteriorated air quality linked with respiratory illnesses cannot be overlooked. The positive correlation between PM2.5 particles and respiratory morbidities in Bhubaneswar signifies that the relatively immune tier II cities also require attention in terms of programme and policies to manage air pollution. Moreover, future studies should target these cities in other similar settings.

## Supporting information

**S1 Checklist. STROBE statement—Checklist of items that should be included in reports of observational studies.**
(DOCX)

## Acknowledgments

We are thankful to the district health officials of Khordha, Medical Officer I/C and staff of participating health centres and Odisha State Pollution Control Board, Bhubaneswar for their valuable support during this study.

## Author Contributions

**Conceptualization:** Abhinav Sinha, Sanghamitra Pati, Jaya Singh Kshatri.

**Data curation:** Abhinav Sinha, Jitendriya Amrit Pritam, Sanghamitra Pati, Jaya Singh Kshatri.

**Formal analysis:** Abhinav Sinha, Jitendriya Amrit Pritam, Hitesh Kumar Jain, Jaya Singh Kshatri.

**Investigation:** Abhinav Sinha, Jitendriya Amrit Pritam, Sidhartha Giri, Sanghamitra Pati, Jaya Singh Kshatri.

**Methodology:** Abhinav Sinha, Jitendriya Amrit Pritam, Sanghamitra Pati, Jaya Singh Kshatri.

**Project administration:** Sidhartha Giri, Sanghamitra Pati, Jaya Singh Kshatri.

**Resources:** Sidhartha Giri, Sanghamitra Pati, Jaya Singh Kshatri.

**Software:** Hitesh Kumar Jain, Sidhartha Giri, Sanghamitra Pati, Jaya Singh Kshatri.

**Supervision:** Sidhartha Giri, Sanghamitra Pati, Jaya Singh Kshatri.

**Validation:** Jitendriya Amrit Pritam, Hitesh Kumar Jain, Sidhartha Giri, Sanghamitra Pati, Jaya Singh Kshatri.

**Visualization:** Abhinav Sinha, Hitesh Kumar Jain, Jaya Singh Kshatri.

**Writing – original draft:** Abhinav Sinha, Jitendriya Amrit Pritam, Jaya Singh Kshatri.

**Writing – review & editing:** Hitesh Kumar Jain, Sidhartha Giri, Sanghamitra Pati.

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
