## [Decision Letter · Decision Letter 0]

6 Jun 2023

PGPH-D-23-00635

Seasonal variations in respiratory morbidity in primary care and its linkages with the quality of air in urban Odisha, India

Dear Dr. Sinha,

Thank you for submitting your manuscript to PLOS Global Public Health. After careful consideration, we feel that it has merit but does not fully meet PLOS Global Public Health’s publication criteria as it currently stands. Therefore, we invite you to submit a revised version of the manuscript that addresses the points raised during the review process.

This manuscript presents interesting data on air quality and respiratory illness in a major Indian city. Such a study would be useful to help expand the evidence base for health effects related to air pollution in India. However, the manuscript needs to be strengthened substantially- both in terms of data analysis and presentation of results as well as discussion of the findings. Please carefully revise the manuscript based on comments from the two reviewers which point to the key weaknesses in the current manuscript. 

In particular, in the discussion section, please discuss the study's findings in the context of available evidence from India- do the results look similar to what we know? Are there important and interesting differences? Furthermore, rather than focusing on policy efforts, please compare and contrast the study's findings with available evidence, and describe any limitations in the analysis.

We look forward to receiving your revised manuscript.

Kind regards,

Pallavi Pant

Academic Editor

Journal Requirements:

1. We have noticed that you have uploaded Supporting Information files, but you have not included a list of legends. Please add a full list of legends for your Supporting Information files after the references list. 

2. In the online submission form, you indicated that "The data analysed during the current study can be obtained from corresponding author on reasonable request". All PLOS journals now require all data underlying the findings described in their manuscript to be freely available to other researchers, either 1. In a public repository, 2. Within the manuscript itself, or 3. Uploaded as supplementary information.

Additional Editor Comments (if provided):

A few additional comments are provided below.

Lines 119-121: The statistic included here (59% of deaths and DALYs) seems very high. Was this meant in the the context of respiratory diseases or the overall burden of disease? Please cross-check and revise the number.

Table 1: Please include age ranges for easy referencing.

Please describe what is meant by "The data analysed during the current study can be obtained

from corresponding author on reasonable request" and also include the name and email address for the corresponding author with the text. 

Reviewers' comments:

Reviewer's Responses to Questions

**Comments to the Author**

1. Does this manuscript meet PLOS Global Public Health’s publication criteria? Is the manuscript technically sound, and do the data support the conclusions? The manuscript must describe methodologically and ethically rigorous research with conclusions that are appropriately drawn based on the data presented.

Reviewer #1: No

Reviewer #2: Yes

2. Has the statistical analysis been performed appropriately and rigorously?

Reviewer #1: No

Reviewer #2: No

3. Have the authors made all data underlying the findings in their manuscript fully available (please refer to the Data Availability Statement at the start of the manuscript PDF file)?

Reviewer #1: Yes

Reviewer #2: Yes

4. Is the manuscript presented in an intelligible fashion and written in standard English?

Reviewer #1: Yes

Reviewer #2: Yes

5. Review Comments to the Author

Reviewer #1: Introduction:

The introduction can be improved upon. The following key areas are missing or not well-articulated.

What is the burden of respiratory diseases?

What is the burden of air pollution?

What is the link between air pollution and respiratory diseases, which consequently prompted the authors to look for the trends in respiratory diseases and related air pollution levels?

Where is the knowledge gap and how is the current study filling that gap?

Line 121- pneumonia is not a chronic respiratory disease.

Methods

The study sites were randomly selected. The authors need to clarify if are all the primary care facilities in the zones of the same level and do they see similar number of patients per month, otherwise, if the health facilities do not have similar background characteristics, there might be bias if a particular type is selected even if it was randomly.

Line 156-157 states that data were collected from January to December in the 2018 and was 157 classified based on the major seasons according to local geo-climatic conditions. Was the classification done at data extraction or only information on month of clinic visit was captured and classification done at analysis?

The details on how information on air pollution was collected are missing. Did they for example consider the daily average or monthly average to decide on the PM2.5 levels for the different seasons?

Data analysis

There is need to clarify on the denominator when calculating the incidence. The cases were derived from the clinic data base- so what was the reference population in this case? And why? In addition, data was collected for non-communicable diseases as well. When extracting data, did the authors consider only newly diagnosed patients or even those who had multiple visits in that year for re-fills/clinical review?

Ethical consideration

Line 181-182 states that authors did not have access to information that could identify participants. In most health systems, the names of the patients are recorded in the register. In this case, I assume that the data was extracted from such register. The authors need to clarify on the local systems on data capture from patients during a clinic visit, how the records are maintained, how they accessed them and how they ensured that all relevant data was accessed. Provide information on who did the data extraction. This information to be included in the description of the study sites and related practices in relation to the study.

Results

Provide detailed information on the air pollution levels during the study period, like monthly average, and show the trend.

The last paragraph in the results section indicates that there was some linkage PM2.5 and incidence of respiratory diagnoses. What type of linkage was this? And what does ‘positive’ mean? Given that the authors have details on diagnoses per season, and air pollution levels ( as I assume), then they can analyse for correlation to enrich the study with such findings.

Discussion

In the second paragraph, the authors note that there was an increase in respiratory diagnoses between 2014 and 2018. Was this in any way related to air pollution levels?

The authors make statements that are not supported by their results. For example, in paragraph 4, there is a statement ‘Furthermore, our findings show, respiratory illnesses increased with a rise in PM2.5 levels in air which also corresponds to the winter season’. Without tests of statistical significance, this statement cannot be validated. The same applies to other statements which mention the linkage without describing the type of linkage and how it was statistically arrived at.

The conclusion of a comprehensive action plan is also not backed by study results

Reviewer #2: First, I would like to appreciate the efforts made by the authors for undertaking this research as it would add to the existing body of literature in the South Asian region on the topic of air quality and its impacts on health. However, the manuscript needs an overhaul before it can reach a publishable stage.

Major Comments:

• In general, the authors need to get the English corrected (sentence format, the sequencing and structure) by sending it to another colleague better in English. Even the authors can use various freely available websites that can support with English language editing.

• Significant concentrated efforts are required to modify the introduction as the essence and storyline is still missing but a better work can be done.

• Are there any adjustments done for any other physical / physiological aspects of the patients concerned for performing this study, should be indicated.

• Figures need special attention.

General Comments:

Abstract:

1. The authors are suggested to rectify the abbreviation used PM2.5 / PM 2.5 should be written as “PM2.5”.

2. Ln 87: This sentence should be shifted above “A positive linkage…….” To Ln 83. Such as the results should start from here and specifics should follow.

Introduction

3. Ln 123-127: They do not fit well in the introduction section, the authors can shift it to the study area, site description and general meteorology section. While describing the study area, the authors could refer previous studies done on the air quality of Bhubaneswar that indicates a seasonal pattern with a winter high and role of long-range transport, while a complete chemical, biological and physical profiling of wintertime particles could support the findings as well as defining the objectives of the current study.

• Seasonal progression of atmospheric particulate matter over an urban coastal region in peninsular India: Role of local meteorology and long-range transport

• Chemical, microstructural, and biological characterization of wintertime PM2.5 during a land campaign study in a coastal city of eastern India

4. Ln 128-138: Needs more modifications in terms of the chronology followed for air pollution mitigation efforts. One major effort is the National Clean Air Program (NCAP) that is not mentioned. Further journal articles summarizing these efforts can also be thoroughly referred, while developing this section.

5. The authors need to bring out the urgency of conducting this study i.e. Bhubaneswar is one of the most rapidly urbanizing cities in India and the projected population growth is also high. Hence, having a better understanding of the health effects would not only help the National efforts, but would be essential for framing state level efforts. While this example is for Bhubaneswar but sever such coastal cities in the South and South-east Asian region with similar or near similar challenges would also be benefitted from such a study.

• https://moef.gov.in/wp-content/uploads/2019/05/NCAP_Report.pdf

• National Clean Air Programme (NCAP) for Indian cities: Review and outlook of clean air action plans

• Monitoring particulate matter in India: recent trends and future outlook

Methodology

6. Ln 156-160: A reference to the climatic classification presented in the study should be given.

7. The data collection section should be subdivided into two parts. First on health data collection and second on air quality data collection.

8. One of the focus of this study is to understand the association of PM2.5 with health, making it essential to provide information on the same. The information should include what was the frequency of PM2.5 data used, how were they averaged or interpreted. From how many stations the data was used and what sort of an instrumentation was used. Something should be mentioned on the QA/QC protocols of collecting the data.

9. Standard references for incidence of morbidity calculation should be used.

Results

10. Pg9 In the first paragraph, the “incidence increased from 53.11 per 1000 year during summer to 54.67 per 1000 person year during rainy season” is this s significant one?

11. Supporting table one with certain descriptive statistics could have been better.

12. Pg10 mentioned on the Pdf has a paragraph “During summer season, … PM2.5 levels.” Is a complete repetition of information provided in above paragraphs. Kindly refrain from doing so.

13. There should be some more discussion keeping in view the figure 2 of this study, as this is the prime topic of the present manuscript.

14. Even the figure 2 doesn’t provide any units

15. Statistical interpretations should be added to quantify the effect of PM2.5 and respiratory morbidities.

Discussions

16. Page no. 11 and 12 as mentioned on the pdf and its last paragraph and first paragraph respectively are extremely vague. Better to remove the same.

17. The discussion in general should also be compared more in terms of the number where they are similar and dissimilar (both) and accordingly reach to a logical reasoning why such a result is seen at Bhubaneswar. To enhance the quality of study and put it in an international context, here studies done within the South and South-East Asian region should be compared and analysed.

6. PLOS authors have the option to publish the peer review history of their article (what does this mean?). If published, this will include your full peer review and any attached files.

**Do you want your identity to be public for this peer review?** For information about this choice, including consent withdrawal, please see our Privacy Policy.

Reviewer #1: No

Reviewer #2: No

---

## [Decision Letter · Decision Letter 1]

17 Oct 2023

PGPH-D-23-00635R1

Seasonal variations in respiratory morbidity in primary care and its linkages with the quality of air in urban Odisha, India

Dear Dr.  Abhinav Sinha

Thank you for submitting your manuscript to PLOS Global Public Health. After careful consideration, we feel that it has merit but does not fully meet PLOS Global Public Health’s publication criteria as it currently stands. Therefore, we invite you to submit a revised version of the manuscript that addresses the points raised during the review process.

We look forward to receiving your revised manuscript.

Kind regards,

Reginald Quansah, Ph.D.

Academic Editor

Journal Requirements:

Additional Editor Comments (if provided):

Reviewers' comments:

Reviewer's Responses to Questions

**Comments to the Author**

1. If the authors have adequately addressed your comments raised in a previous round of review and you feel that this manuscript is now acceptable for publication, you may indicate that here to bypass the “Comments to the Author” section, enter your conflict of interest statement in the “Confidential to Editor” section, and submit your "Accept" recommendation.

Reviewer #1: All comments have been addressed

Reviewer #2: All comments have been addressed

2. Does this manuscript meet PLOS Global Public Health’s publication criteria? Is the manuscript technically sound, and do the data support the conclusions? The manuscript must describe methodologically and ethically rigorous research with conclusions that are appropriately drawn based on the data presented.

Reviewer #1: Yes

Reviewer #2: Yes

3. Has the statistical analysis been performed appropriately and rigorously?

Reviewer #1: Yes

Reviewer #2: Yes

4. Have the authors made all data underlying the findings in their manuscript fully available (please refer to the Data Availability Statement at the start of the manuscript PDF file)?

Reviewer #1: Yes

Reviewer #2: (No Response)

5. Is the manuscript presented in an intelligible fashion and written in standard English?

Reviewer #1: Yes

Reviewer #2: (No Response)

6. Review Comments to the Author

Reviewer #1: Line 120-122: The statement about the burden of asthma and COPD better suited at the end of the next paragraph where information on disease burden is provided.

Line 171-174: the season that is being compared to the pre-monsoon is not stated.

Line 210-212: This information is about health data, but it has been put under the sub-heading of air pollution. This needs to be revised.

In paragraph 4 of the discussions, the authors seem to authoritatively suggest a causal relationship between air pollution and respiratory morbidity as indicated in the sentence ‘It is evident that respiratory morbidities are evidently linked with the air pollution which needs attention at all levels’. However, it is important to note that this was a chart review where causal relationships are difficult to prove, and that there are many confounders of respiratory morbidity that were not controlled for. It might be better to use clear epidemiological terms like association or correlation throughout the documents rather than ‘linkage’ which is not very specific.

Reviewer #2: The authors have made a decent attempt to address the comments. Some minor comments (appended below) need to be addressed before it can be published.

Comments

• Ln 200-212: Please mention the time of air quality data used. Something should be mentioned on the QA/QC protocols of collecting the data. Kindly talk to the PCB officials and mention it accordingly.

• Figures 3 need special attention.

7. PLOS authors have the option to publish the peer review history of their article (what does this mean?). If published, this will include your full peer review and any attached files.

**Do you want your identity to be public for this peer review?** For information about this choice, including consent withdrawal, please see our Privacy Policy.

Reviewer #1: No

Reviewer #2: **Yes: **Parth Sarathi Mahapatra

---

## [Editor Report · Decision Letter 2]

11 Jan 2024

Seasonal variations in respiratory morbidity in primary care and its correlation with the quality of air in urban Odisha, India

PGPH-D-23-00635R2

Dear Dr. Jayasingh Kshatri,

We are pleased to inform you that your manuscript 'Seasonal variations in respiratory morbidity in primary care and its correlation with the quality of air in urban Odisha, India' has been provisionally accepted for publication in PLOS Global Public Health.

Best regards,

Reginald Quansah, Ph.D.

Academic Editor